# Synthesis of a New Phosphonate-Based Sorbent and Characterization of Its Interactions with Lanthanum (III) and Terbium (III)

**DOI:** 10.3390/polym13091513

**Published:** 2021-05-08

**Authors:** Yuezhou Wei, Khalid A. M. Salih, Mohammed F. Hamza, Toyohisa Fujita, Enrique Rodríguez-Castellón, Eric Guibal

**Affiliations:** 1Guangxi Key Laboratory of Processing for Non-ferrous Metals and Featured Materials, School of Resources, Environment and Materials, Guangxi University, Nanning 530004, China; yzwei@gxu.edu.cn (Y.W.); Immortaltiger7@gmail.com (K.A.M.S.); fujitatoyohisa@gxu.edu.cn (T.F.); 2Guangdong Institute of Rare Metals, Guangdong Academy of Science, Guangzhou 510651, China; 3Nuclear Materials Authority, El-Maadi, Cairo POB 530, Egypt; 4Departamento de Química Inorgánica, Facultad de Ciencias, Universidad de Málaga, 29016 Málaga, Spain; 5Polymers Composites & Hybrids (PCH), IMT—Mines Ales, 30100 Alès, France; eric.guibal@mines-ales.fr

**Keywords:** phosphonate-functionalization, rare earth elements, enhanced sorption capacities, FTIR and XPS characterization, interpretation of binding mechanisms, algal/polyethylenimine beads

## Abstract

High-tech applications require increasing amounts of rare earth elements (REE). Their recovery from low-grade minerals and their recycling from secondary sources (as waste materials) are of critical importance. There is increasing attention paid to the development of new sorbents for REE recovery from dilute solutions. A new generation of composite sorbents based on brown algal biomass (alginate) and polyethylenimine (PEI) was recently developed (A_L_PEI hydrogel beads). The phosphorylation of the beads strongly improves the affinity of the sorbents for REEs (such as La and Tb): by 4.5 to 6.9 times compared with raw beads. The synthesis procedure (epicholorhydrin-activation, phosphorylation and de-esterification) is investigated by XPS and FTIR for characterizing the grafting route but also for interpreting the binding mechanism (contribution of N-bearing from PEI, O-bearing from alginate and P-bearing groups). Metal ions can be readily eluted using an acidic calcium chloride solution, which regenerates the sorbent: the FTIR spectra are hardly changed after five successive cycles of sorption and desorption. The materials are also characterized by elemental, textural and thermogravimetric analyses. The phosphorylation of A_L_PEI beads by this new method opens promising perspectives for the recovery of these strategic metals from mild acid solutions (i.e., pH ~ 4).

## 1. Introduction

The development of high-tech devices causes an increasing demand for rare earth elements (REEs) [1]. This pressure may explain the incentive politics defended by international institutions for recovering these strategic metals from secondary resources such as low-grade minerals [2] and phosphate rocks [3], but also from waste industrial materials [4,5,6]. Hydrometallurgical processes are commonly used for recovering these metals from solid wastes and ores through acid leaching processes [7,8,9,10,11].

Sorption processes have been widely designed for the recovery of rare earth elements, including functionalized silica [12], carbon-based sorbents [13], chemically modified membranes [14], metal organic framework [15], ion-exchange resins and chelating resins [16,17].

Solvent extraction is a standard process currently employed for the removal of REEs from acidic leachates [18,19,20,21], especially for solutions containing high metal concentrations. Most of these highly efficient extractants are based on phosphorus reagents such as alkyl phosphate [19,22,23,24,25,26], alkyl phosphoric acid [27,28,29], alkyl phosphine oxide [30,31], phosphonium ionic liquids [32,33] and organophosphonic extractants [34,35,36,37]. The strong affinity of phosphorus-based extractants for rare earth easily explains that a strong effort has been made for designing synthetic resins bearing phosphorus-based reactive groups such as Tulsion CH-96 and T-PAR resins [38,39], Tulsion CH-93 [34], Purolite S957 and Diphonix [40] or metal-organic frameworks [41]. Heres et al. [42] compared a series of resins bearing phosphonic reactive groups (multifunctional resins) for the extraction of REEs from phosphoric acid and they reported that aminophosphonic IRC-747 and aminomethylphosphonic TP-260 resins are the most promising. The chemical environment of phosphonic groups turned out to be particularly important for the sorption performances of this type of resin in the recovery of REEs but also for other metal ions such as U(VI) [43,44].

Recently, a new type of support has been designed based on the one-pot synthesis of a composite associating alginate extracted from brown algae (*Laminaria digitata* brown alga) and polyethylenimine (PEI). The interpenetrating network created by the interaction between carboxylic acid groups of alginate and amine groups of PEI, completed by the ionotropic gelation of carboxylic groups of alginate with calcium chloride and the crosslinking of amine groups with glutaraldehyde allows producing very stable beads. In some cases, it may be useful to introduce amounts of alginate in the synthesis procedure to increase the stability of the composite. Alginate has been used for incorporating and stabilizing magnetic nanoparticles for the synthesis of composite sorbents, and applied for rare earth elements [45,46]. These materials are multifunctional bearing carboxylic groups and amine groups with a high affinity for a wide range of metal ions (depending on the pH): metal anions in acidic solutions and metal cations in near-neutral solutions. The presence of these reactive groups also allows for readily functionalizing the sorbent by activation of amine groups. Hence, a series of sorbents functionalized with amidoxime groups [44], quaternary ammonium groups [47], sulfonic groups [48] and phosphoryl groups [49] have been designed. Studies confirm the contribution of several groups on multifunctional sorbents in the recovery of selected metal ions (uranyl ions, metalloid anions, and members of the REE family).

The current work describes a new derivative of algal/PEI beads (A_L_PEI, with addition of alginate) obtained by activating the support by epichlorohydrin grafting, followed by phosphorylation and de-esterification (to make free the phosphonic acid groups, in addition to other reactive functions: carboxylic and amine groups, POH-A_L_PEI). This sorbent is tested for the sorption of two REEs: La (III) (as a representative of light REEs, LREEs) and Tb (III) (heavy REEs, HREEs). This specific part of the study focuses on (a) the characterization of the sorbent (in the different stages of the synthesis procedure), and (b) the identification of the binding mechanisms by spectral analysis: Fourier-transform infrared and X-ray photoelectron spectroscopies. The effect of the pH is also considered as a complementary contribution to the interpretation of binding mechanism.

## 2. Materials and Methods

### 2.1. Materials

Algal biomass (*Laminaria digitata*) was kindly supplied by Setalg (Pleubian, France). After grinding, algal biomass was sieved and particles below 250 µm were used for preparing the raw beads. Branched polyethylenemine (PEI, 50%, w/w in water), and glutaraldehyde (GA, 50%, w/w in water) were supplied by Sigma-Aldrich (Taufkirchen, Germany). Polyethylene glycol diglycidyl ether (MW_n_: ~500 g mol^−1^), triethyl phosphite, epichlorohydrin and trimethylsilyl bromide were purchased by Shanghai Makclin Biochemical Co., Ltd. (Shanghai, China). Dichloromethane was supplied by Xilong Scientific Co., Ltd., (Guangdong, China). Ethanol absolute and sodium hydroxide were purchased from Guangdong Guanghua Sci-Tech (Guangzhou, China). Lanthanum (III) sulfate and terbium (III) sulfate were supplied by National Engineering Research Centre of Rare Earth Metallurgy and Functional Materials Co., Ltd., Shijiazhuang, China. Silicon standard solution (1000 ppm, used as the source for Si) was supplied by Guobiao Inspection and Certification Co. Ltd. (Huairou District, Beijing, China). The other reagents were Prolabo products (VWR, Radnor, PA, USA).

### 2.2. Synthesis of Functionalized Sorbent (POH-A_L_PEI)

#### 2.2.1. Synthesis of Algal/PEI Beads (A_L_PEI)

Composite homogeneous algal/PEI beads were prepared by a three-step procedure: (a) partial alginate extraction from algal biomass, (b) mixing with PEI solution and (c) ionotropic gelation. Finally, the beads were freeze-dried (−52 °C, 0.1 mbar) for two days. After grinding and sieving algal biomass (*L. digitata*), the fraction below 250 µm (30 g) was dispersed into a 1% (w/w) Na_2_CO_3_ solution (800 mL). The suspension was mixed for 24 h at a temperature of 50 °C for processing the partial extraction of alginate from algal biomass. Then, the suspension was mixed with 5 mL of PEI (50%, w/w); the suspension was dropped into 2 L of CaCl_2_ solution (1%, w/w), containing 5 mL of GA (50 w/w). The ionotropic gelation of alginate extracted from algal biomass operates through the interaction of carboxylate groups with calcium ions. The interaction of protonated amine groups of PEI with the carboxylate groups and crosslinking of amine groups from PEI moieties with carbonyl groups from GA also contribute to stabilize the beads (multiple interpenetrating network: alginate/PEI and alginate/Ca (II)).

#### 2.2.2. Epichlorohydrin-Activation of A_L_PEI Beads (Cl-A_L_PEI)

Five grams of A_L_PEI beads were dropped into 100 mL ethanol/water solution (1:1, *v/v*). Three grams of poly(ethyleneglycol) diglycidyl ether (PEGDE, for improving the strength of the beads and enhancing their stability) were added to the mixture under stirring for 5 h at 80 °C (reactor equipped with a condenser).

Epichlorohydrin is a mediator frequently used in the grafting of functional groups onto polymer and biopolymer supports [50]. The beads were then transferred to a reactor containing 200 mL ethanol/water solution (1:1, *v/v*), in which a volume of 45 mL of epichlorohydrin was dropped wisely under gentle stirring. The suspension was refluxed for 3 h at 80–85 °C. The beads were filtered, washed with ethanol, and dried under vacuum to produce activated beads (spacer Cl-bearing arms). 

#### 2.2.3. Phosphorylation of Activated Beads (P*-A_L_PEI)

The activated beads (Cl-A_L_PEI) were dropped into 100 mL of triethyl phosphite. The reaction mixture was heated at 130 °C for 24 h to produce phosphorylated A_L_PEI beads (P*-A_L_PEI). The beads were filtered and washed several times with hot water and ethanol then dried under vacuum overnight.

#### 2.2.4. De-Esterification of P*-A_L_PEI Beads (POH-A_L_PEI)

The phosphorylated beads (P*-A_L_PEI) were soaked into 75 mL of dichloromethane under stirring for 24 h at room temperature. Trimethyl silyl bromide (10 g) was added to the suspension under nitrogen and stirring at 30–35 °C for 24 h to produce phosphonic acid-functionalized beads (POH-A_L_PEI). The beads were washed several times with hot water to remove the excess of unreacted bromide derivative, before being rinsed with acetone and dried in vacuum overnight.

Scheme 1 summarizes the different steps of the synthesis procedure.

### 2.3. Characterization of Materials

An ESCALAB 250XI+ instrument (Thermo Fischer Scientific, Inc., Waltham, MA, USA) was operated for collecting XPS spectra of the different materials (synthesis procedure, before and after metal sorption). The excitation source was monochromatic Al K_α_ X-ray radiation (1486.6 eV) applied on a 500 µm spot size. The pressure of the analytical chamber was less than 10^−8^ mbar. The analyzer used a 20 eV pass energy for the acquisition of high-resolution core level spectra (50 eV for full spectrum analysis). The BE calibration was performed using C 1*s* at 284.8 eV for C_adv_, C-C- and C=C. The energy resolution on Ag 3*d*_5/2_ was 0.45 eV and 0.82 eV for C 1*s*. FTIR spectra were acquired on dried samples (dispersed into KBr discs) using an IRTracer-100 (Shimadzu, Tokyo, Japan). 

The morphology observation and the semi-quantitative surface analyses were obtained on a Phenom ProX scanning electron microscope (SEM, Thermo Fisher Scientific, Eindhoven, Netherlands) coupled with EDX facilities. The textural properties of the sorbents have been qualified using the BJH method and a Micromeritics TrisStar II (Norcross, GA, USA). The pre-treatment of the samples consisted of their degassing at 100 °C for 12 h. A Netzsch STA 449 F3 Jupiter thermogravimeter was used for TGA analysis, with a temperature ramp of 10 °C/min (under oxygen or air atmosphere) (NETZSCH-Gerätebau GmbH, Selb, Germany). The elemental composition of the sorbents was obtained on a Vario EL cube element analyzer (Elementar Analysensysteme GmbH, Langenselbold, Germany). The pH_PZC_ was determined using the pH-drift method [51].

### 2.4. Metal Sorption

The loading of the sorbent with metal ions (La(III) and Tb(III)) was operated in a batch system by contact of the beads (m, g) with a volume (V, L) of solution containing 50 mg metal L^−1^ (i.e., C_0_: 0.375 mmol La L^−1^ and 0.314 mmol Tb L^−1^) at pH_0_: 4, room temperature for 24 h. Final pH was recorded: 4.40–4.45 for La(III) and 4.3–4.4 for Tb(III). The investigation of the effect of the pH was carried out using the same procedure, except that the initial pH was varied in the range pH_0_: 1–5; the sorbent dose, SD (g L^−1^, SD = m/V). The final pH was monitored, and the residual concentration (C_eq_, mmol L^−1^) was determined using an ICP-AES spectrometer (inductively coupled plasma atomic emission spectrometer, ICPS-7510 Shimadzu, Tokyo, Japan). The amount of metal bound to the sorbent (q_eq_, mmol g^−1^) was deduced from the mass balance equation: q_eq_ = (C_0_-C_eq_) ×V/m. The distribution ratio (D, L g^−1)^ is calculated by D = q_eq_/C_eq_. The FTIR spectra of materials exposed to five successive cycles of sorption/desorption for evaluating the stability of the sorbent. The eluent was an acidic calcium chloride solution (i.e., 0.2 M HCl/0.5 M CaCl_2_). In the investigation of sorbent recycling, a rinsing (with demineralized water) was systematically performed between each step.

## 3. Results and Discussion

### 3.1. Physical Characterization of Sorbents

#### 3.1.1. Morphology and Textural Characterization

Figure 1 shows that the functionalized sorbent (POH-A_L_PEI beads) is characterized by a roughly spherical shape; although some deformations occurred (probably associated to the drying steps of the hydrogels). The color of the beads has become lighter after metal sorption.

Appendix A shows optical micrographs of the materials: the sorbents are roughly spherical, and the average bead size progressively decreases with the chemical modification of A_L_PEI (from 2.7 mm to 2.07 mm). The surface is smoothed but uneven. This is confirmed by the SEM micrographs of the surface of the beads at the different stages of the synthesis (Appendix A). On POH-A_L_PEI beads, micro-cracks and folds appear on the surface of the sorbent. The SEM microphotographs of the crosscut section show the highly macroporous internal structure (laminated structure characterized by sheets and scaffolds; 50–200 µm size cell opening), especially for functionalized materials. Appendix A shows the semi-quantitative EDX analyses of surface and crosscut sections. The activation of the raw material (Cl-A_L_PEI) is confirmed by the appearance of Cl (4.1–4.7%, weight fraction), the disappearance of Cl after phosphorylation (P*-A_L_PEI) and de-esterification (POH-A_L_PEI) confirms the synthesis mechanism. The semi-quantitative analysis is roughly the same for surface and crosscut sections. After metal sorption, as expected, La and Tb appear (at levels close to 2.6–3.1%, weight fraction) (see below); the weight fraction of S increases from 0.2–0.3% (sorbent) to 1.7–2% after metal binding. This is consistent with the observations from the FTIR and XPS analyses.

SEM pictures show that the materials can be considered heterogeneous with a tight external skin and a highly macro-porous internal compartment. The “compact” thin external layers proceeds from the ionotropic gelation of the beads while the macroporous cavities are associated with the freeze-drying procedure of the hydrogel network. The functionalization tends to block and/or coat the residual micro-porosity initially present in raw beads (A_L_PEI).

#### 3.1.2. Thermogravimetric Analysis

Appendix A compares the profiles of thermal degradation (TGA and DrTG) for A_L_PEI and POH-A_L_PEI beads (and intermediary synthesis products). The TGA profile of A_L_PEI is marked by four degradation steps, corresponding to:Water release (below 198 °C).Degradation of carbohydrate ring, polyethylenimine depolymerization (in the range 198–481 °C, maximum DrTG at 319.1 °C).Degradation of carbon-based backbone leading to char formation (in the range 184–639 °C, DrTG_max_ at 500.2 °C).And thermal degradation of the char (DrTG_max_ at 500.2 °C). Total weight loss reaches 82.9% at 909 °C.

The chemically modified materials are characterized by smoother TGA profiles, where only 3 steps are identified; however, the DrTG profiles show an increased number of transitions (see Table associated with Appendix A), probably associated to the greater diversity of functional groups present on the materials. It is noteworthy that the thermal stability of the Cl-A_L_PEI and P*-A_L_PEI materials is increased compared with the raw material: (a) the weight loss does not exceed 76.7 and 78.9%, respectively, and (b) the weight loss curves are shifted toward higher temperatures (~500 °C, ~549 °C and ~600 °C for A_L_PEI, Cl-A_L_PEI and P*-A_L_PEI, respectively). The incorporation of phosphorus compounds is well-known for improving the fire-properties of materials [52]. On the opposite hand, after de-esterification, the thermal degradation profile roughly turns back to the profile of A_L_PEI below 400 °C, while the degradation is enhanced at higher temperatures: the total weight loss reaches 96.2% at 909 °C. This diversity of degradation profiles confirms the substantial chemical modifications of the functionalized polymers in the different stages of the synthesis.

#### 3.1.3. Elemental Analysis and pH_PZC_

Another proof of the chemical modifications of the A_L_PEI beads is provided by the comparison of the elemental compositions, which is summarized in Appendix A. The progressive increase of the C fraction with chemical modification is consistent with the grafting of new organic functions on the composite backbone, while de-esterification slightly decreases the C content (cleavage of ethyl groups). The N content decreases with the sorbent functionalization (from 3.48 to 2.78 mmol N g^−1^): the grafting of supplementary organic functions logically reduces the relative amount of nitrogen. The variation in the O content follows a reciprocal tend: the O content progressively increases in the different stages of the synthesis, from 21.80 to 27.03 mmol O g^−1^. This is consistent with the reagents used for activation (Cl-A_L_PEI) and for phosphorylation (3 O per activated amine group). Efficient phosphorylation is demonstrated by quantification of P (reaching up to 2.1 mmol P g^−1^). This phosphorylation on activated amine groups is highly efficient: the N content in Cl-A_L_PEI is close to 3 mmol g^−1^, to be compared with the final P content in the sorbent. This means that the grafting is close to 70%; this is remarkable taking into account that (primary, secondary and tertiary, P-S-T) amine groups may have different reactivity and accessibility in branched PEI.

Appendix A compares the pH_PZC_ of raw and functionalized beads. Chemical modification of A_L_PEI significantly shifts the pH_PZC_ of the material (from 4.29 to 6.82). The raw material bears amine groups (P-S-T; pK_a_: 4.5, 6.7 and 11.6, respectively) from PEI and carboxylic groups (mannuronic and guluronic acid; pK_a_: 3.38 and 3.65, respectively) from alginate. By grafting phosphonic groups the acid-base properties are substantially modified, and the sorbent is protonated in acidic solutions (including close to a neutral pH). Kolodynska et al. [53] compared the sorption properties of a series of chelating resins for lanthanum sorption and reported pH_PZC_ values of aminophosphonic-based resins in the range 8.13–9.93. Glowinska and Trochimczuk [54] reminded that the two pK_a_ of phosphonic groups are close to 1.3 and 6.70. These properties are critical for defining the affinity of sorbents for target metal ions (attraction/repulsion) and sorption mechanisms (chelation vs. ion-exchange/electrostatic attraction).

### 3.2. Synthesis of POH-A_L_PEI Beads—Chemical Characterization

The different materials elaborated in the suggested route for sorbent synthesis (as reported in Scheme 1) are analyzed by FTIR and XPS spectroscopies to identify the reactive groups and how they are affected by the successive reactions. 

#### 3.2.1. FTIR Spectroscopy

Figure 2 compares the FTIR spectra of the raw material (A_L_PEI) and the sorbent (POH-A_L_PEI), including intermediary products. Focused windows (in specific wavenumber ranges) are reported in Appendix A. Brown algal biomass (*L. digitata*) contains an important alginate: based on the literature, this fraction may reach up to 50% [55], although this composition may vary with location and season [56,57]. This fraction of alginate is partially extracted by sodium carbonate with heating and may be available for further reaction with other constituents of the sorbent. In addition, the biomass contains other carbohydrates (such as fucoidan) or sugars (glucose and mannitol) and proteins. 

The incorporation and reaction of protonated amine groups of PEI with carboxylic acid of alginate contributes to the structuration of the hydrogel, which is reinforced by gelation of the alginate with calcium chloride. This means that a wide variety of functional groups may be present on the support (A_L_PEI), e.g., mainly, carboxylic groups (alginate, proteins), sulfonic groups (from fucoidans), amine groups (from PEI, proteins; primary, secondary and tertiary amines) and hydroxyl groups. This may explain the broad bands observed in the FTIR spectra for A_L_PEI and its derivatives: the superposition of spectral domains makes the interpretation and identification of peaks that may overlap and convolve complex. Different important spectral regions can be identified in relation with the chemical groups present in the materials [58]. Appendix A reports the assignments of the main peaks.

Region 3800–2200 cm^−1^

In this wavenumber range, a poorly resolved broad band can be identified by a peak at ~3450 cm^−1^ and a shoulder at ~3250 cm^−1^, corresponding to the convolution of different stretching vibrations of -OH and -NH. In addition, stretching vibrations of -CH bonds are appearing in the region 2980–2850 cm^−1^ confirming the expected presence of linear aliphatic chains. It is noteworthy that additional weak peaks can be observed at lower wavenumber (i.e., around 2762 and 2739 cm^−1^), which could be assigned to methylamino groups. Two peaks are identified at ~2430 and ~2400 cm^−1^, which may be attributed to carbonate and/or CO_2_ absorption. Actually, the four materials show very comparable FTIR patterns. Chemical changes do not appear clearly in this wavenumber range, because of the superimposition of different bands present in the different reagents.

Region 1800–1200 cm^−1^

In this region, the sorbents also show very similar patterns. A first well-resolved band is observed at 1767 cm^−1^, which is usually assigned to C=O stretching vibration in carboxylic groups. Its relative intensity decreases after reaction with epichlorohydrin and PEGDE (Cl-A_L_PEI). The intensity continues to decrease with phosphorylation (P*-A_L_PEI), which is accompanied by the appearance of a weak peak at 1726 cm^−1^ (attributed to C=O stretching in ester groups). This additional weak peak disappears after de-esterification (POH-A_L_PEI), due to the increase in the contribution of the OH stretching vibration. The second broad peak is observed in the wavenumber range 1632–1622 cm^−1^; this large band may be assigned to C=O asymmetric stretching in (COO^−^), (-N-H) bending vibration (for primary and secondary amines) and/or open-chain imino groups (-C=N-). The strongest peak is identified for the four materials at 1382 ± 1 cm^−1^. This peak is broad as shown by the full width at half maximum (FWHM) that exceeds 100 cm^−1^ and corresponds to the superposition of different signals corresponding to contributions of carboxylate (symmetric stretching, 1420–1300 cm^−1^), -OH bending (for carboxylic acid and aldehyde), -C-H bending, C-N stretching and S=O stretching (for sulfonic groups of fucoidans). The activation of A_L_PEI is followed by the appearance of a new peak at 1213 cm^−1^. This peak is a tracer of chloromethylation of amine groups (associated with the action of PEGDE); Kuo et al. [59] investigated the epoxidation of bark with epichlorohydrin and reported the appearance of a specific epoxide band in this region. The peak disappears almost completely after phosphorylation. This is a confirmation of the efficiency of epichlorohydrin in promoting the quantitative phosphorylation of the support.

Region 1200–500 cm^−1^

The spectrum of A_L_PEI shows weak peaks in the 1170–1050 cm^−1^ range that correspond to C-C, C-O-C and C-O (in alcohols) stretching vibrations (associated with carbohydrate ring), and C-N stretching. After epichlorohydrin-activation (including PEGDE grafting), the spectrum of Cl-A_L_PEI shows a series of peaks at 1080 cm^−1^ (C-O and C-C stretching), 1055 cm^−1^ (C-N stretching), 1001 cm^−1^ (C-O-C stretching), 949 cm^−1^ (epoxide asymmetric stretching, [60]), and 827 cm^−1^ (CH_2_-Cl stretching). The relative intensities of these bands decrease after phosphorylation (P*-A_L_PEI), while the sharp peak at 827 cm^−1^ disappears: the grafting operates quantitatively on the pending CH_2_-Cl groups.

Zenobi et al. reported the effect of pH on the FTIR spectra of different phosphonic acids in solution [61], and after sorption onto boehmite [62]: significant shifts are observed when varying pH and these changes are completed with additional peaks after adsorption (double impact of pH and interaction/chemical environment). The phosphorylation of Cl-A_L_PEI (i.e., P*-A_L_PEI) is characterized by the appearance of a series of weak peaks at 1107 cm^−1^ (organic phosphates P=O stretch), 1018 cm^−1^, 942 cm^−1^ and 527 cm^−1^ (symmetric stretching of P-O-C), and 1050–990 cm^−1^ (P-O-C stretching) [61]. After de-esterification (i.e., POH-A_L_PEI), the intensity of shoulder at around ~1180 cm^−1^ decreases (stretching of P=O band, [63]); the bands at 1018 cm^−1^ and 527 cm^−1^ are shifted to 1055 and 571 cm^−1^, respectively. A doublet also appears at 837 and 831 cm^−1^ (probably associated with P-O-C deformation, ~780 mn^−1^ according to Illy et al. [63]). 

The change in the environment of phosphorus in phosphonate functional groups significantly modified the FTIR spectrum in this region, as a confirmation of the de-esterification reaction.

#### 3.2.2. XPS Spectroscopy

The characterization of A_L_PEI functionalization is completed by XPS analysis of the different intermediary compounds. Figure 3 shows the survey spectra of the studied samples (where selected binding energy ranges are highlighted; completed by selected HRES signals, appearing in Appendix A). 

The chemical modifications are clearly shown when comparing the profiles of the signals and their deconvolutions. Figure 4 shows the substantial changes of the C 1*s* core level spectra along the successive grafting steps. With the epichlorohydrin activation, the relative intensities of the deconvoluted signals weakly change (limited shifts).

The phosphorylation drastically increases the relative intensities of C-C and C=C against those of C-OH, C-O-C, C=O (amide), and carboxyl. The de-esterification step restores the C 1*s* core level spectrum (similar to the epichlorohydrin-activated material (due to the change in the chemical environment of phosphoryl group). Some changes are also observed on O 1*s* core level spectra (Figure 5); apart from the changes in intensity and weak shifts, the most relevant differences are observed for P*-A_L_PEI and POH-A_L_PEI (phosphorylated materials) with the disappearance of C-O (carboxylate) and C=O (amide) contributions. For N 1*s* core level (Figure 6), the modification of A_L_PEI is followed by a reinforcement of the contribution of alkyl ammonium signal. In addition, the de-esterification leads to the appearance of a new signal, assigned to N (amide) at BE: 400.6 eV. The de-esterification of P*-A_L_PEI, maintains the shape of the P 2*p* core level spectrum; however, a shift towards lower BEs is observed (Figure 7). Appendix A shows the binding energies (BEs, eV), atomic fraction (%) and possible assignments of the deconvoluted peaks. The successive modifications of the raw beads lead to substantial changes in C 1*s*, O1*s* and N 1*s* core level spectra with modifications in the relative contributions and appearance/disappearance of specific functional groups. For example, in the case of C 1*s*, the activation of the support hardly changes the HRES XPS profile, with only a weak decrease in the relative contribution of amide group. It is possible to assign the signal at 286.5 eV to different contributions, including to that of the C-Cl covalent bond [64]. This C-Cl bond is confirmed by the Cl 2*p*_3/2_ contribution at 199.9 eV for Cl-A_L_PEI, in addition to the C-Cl covalent bonding. In contrast, after phosphorylation, the profile completely changes with a significant reinforcement of the peak at ~284.8 eV (C-C and C=C functional groups) and C-P of the phosphonic group [65].

### 3.3. Characterization of the Interactions of POH-A_L_PEI with La(III) and Tb(III) Metal Ions

#### 3.3.1. FTIR Spectroscopy 

For A_L_PEI, the most significant changes appearing with metal sorption are identified in the regions (Figure 8, Appendix A): ~1753 cm^−1^—the C=O stretching band disappears after La (III) and Tb (III) sorption;And 1440–1370 cm^−1^—the major peak is shifted from 1382 cm^−1^ to ~1430 cm^−1^. Actually, the FTIR spectra show a broad band, which results from the convolution of different signals; after metal binding, relative contributions of carboxyl and amine groups are modified: these groups are involved in metal binding.

It is noteworthy that after metal desorption the peaks are restored: the 1753 cm^−1^ peak reappears, while the ~1430 cm^−1^ band is shifted back to ~1382 cm^−1^. This means that the sorbent can be readily regenerated with limited degradation of the material (see below the confirmation of performance stability through recycling tests).

In the case of POH-A_L_PEI, the peaks relative to carboxyl and amine groups appear to be less affected by metal uptake (~1753 cm^−1^ and ~1440–1370 cm^−1^): a slight enlargement of the broad band may be identified at ~1350 cm^−1^. More significant variations appear around 1115 cm^−1^ (P-O stretching vibration), 1057 cm^−1^ (C-O and C-N stretching vibration), in the range 838–808 cm^−1^ (P-O-C stretching vibration), at 617 cm^−1^ (sulfate ion) and 571 cm^−1^ (P-O-C stretching vibration). The sorption of La (III) and Tb (III) is followed by the disappearance of the P-O-C vibration (at 838–808 cm^−1^ and 571 cm^−1^). After metal elution, the peak at 838–808 cm^−1^ reappears (contrary to the peak at 571 cm^−1^: the sorption/desorption cycle durably affects the environment of this reactive group). The appearance of the peak at 1115 cm^−1^ (intensity increase) confirms that metal sorption reversibly affects the environment of phosphonic groups. The weakening of the peak at 1057 cm^−1^ shows that the amine and carboxyl groups may contribute to metal binding; however, the weak density of these reactive groups after functionalization of the support alters the visibility of this peak. The peak at 617 cm^−1^ (attributed to sulfate ion) indicates that the metals are probably bound as metal-sulfate species (or at least that sulfate anions are bound on the sorbents); the peaks disappear after metal desorption. 

#### 3.3.2. XPS Spectroscopy

XPS spectroscopy can be used not only for the characterization of the sorbent but also for the identification of binding mechanisms [66,67]. The sorption mechanisms can be also approached by the variations in the XPS spectra of the sorbents exposed to La (III) and Tb(III) solutions for selected signals (Figure 9, survey spectra, Figure 10 and Figure 11 HRES XPS spectra, and Appendix A).

The HRES spectra for selected elements are reported in Appendix A. The C 1*s* signal is poorly affected by metal binding in terms of BEs and AFs. The signal O 1*s* shows more modifications: little shifts in the BEs (−∆ BEs for both carboxylate/C=O (amide) groups and O-P phosphonic groups) and decrease in relevant AFs. The deconvolution of the global signal reveals after metal sorption the appearance of supplementary O-H (from H_2_O) on the sorbent: the metal ions may be sorbed under their hydrated form (solvated form). In addition, the appearance of S 2*p* signal confirms the FTIR observations: appearance of sulfate by direct binding and/or binding of metal-sulfate species. The comparison of deconvoluted signals for N 1*s* signal shows that metal binding is followed by a decrease of the relative contribution of alkyl ammonium signal (in favor of amine and amide signals): the environment of amine groups is probably affected by metal bonds, probably associated with charge neutralization of protonated amine groups with anionic metal species. The P 2*p* signal is sifted toward higher binding energy values after Tb (III) sorption (∆ BEs: +0.4–0.5 eV) and even more for La (III) binding (∆ BEs: +1.5–1.6 eV). The sorption of these REEs involves a contribution of phosphonic groups, consistently with the conclusions raised for FTIR study. The presence of lanthanum is identified by the doublet La 3*d*_5/2_–La 3*d*_3/2_ at ~835 and ~851 eV, respectively, (including relative multiplets with ∆ BEs: ~3.2–3.4 eV). In the case of Tb (III), the deconvolution is more difficult due to poorly resolved signals. More specifically, the analysis of the Tb 4*d*_5/2_ signal tends to indicate that two forms of Tb may coexist at ~149 eV and 152–151 eV; the REE may coexist as bound Tb (III) but also under its oxidized form, i.e., Tb (IV), respectively [68,69].

#### 3.3.3. Effect of pH on La (III) and Tb (III) Sorption onto POH-A_L_PEI

Figure 12 compares the sorption properties of La (III) and Tb (III) for raw and functionalized sorbents. The comparison of the profiles clearly demonstrates the strong improvement in sorption properties associated with the phosphorylation of the sorbent. In the case of A_L_PEI, the sorption capacity does not exceed 0.104 mmol La g^−1^ and 0.055 mmol Tb g^−1^ in the pH range pH_0_: 1–5. On the other hand, the phosphorylation of the beads allows increasing sorption capacities to 0.465 mmol La g^−1^ and 0.373 mmol Tb g^−1^, under the same experimental conditions. This significant increase (×4.5 for La and ×6.8 for Tb) may be explained by the increase in the density of reactive groups, their higher affinity for phosphonic groups and the more favorable acid–base characteristics (see pH_PZC_). As the pH increases, the sorption capacities strongly increase for POH-A_L_PEI (especially above pH_0_ 2), while for A_L_PEI, the effect of pH is quickly leveled.

Appendix A reports the speciation diagrams of lanthanum and terbium under the experimental conditions selected for the study of the pH effect. The predominant species are LaSO_4_^+^ and TbSO_4_^+^ below pH 4; between pH 4 and 6, these species co-exist, at comparable levels, with free La^3+^ and Tb^3+^. The anionic metal disulfate species (La (SO_4_)_2_^−^ and Tb (SO_4_)_2_^−^) exist only below pH 3 and never exceed 15–20%. Throughout the pH range investigated in this study, the largely predominant species are cationic. Considering that the pH_PZC_ of A_L_PEI is close to 4.29, the protonation of the sorbent may explain repulsion effects that limit sorption efficiency. Charge repulsion decreases with increasing pH and enhances metal sorption. However, even at pH 2, the sorption is not negligible. In addition, despite the negative charge of the sorbent at pH 5, the sorption does not increase dramatically. This probably means that the ion-exchange mechanism is not the main reaction pathway. In the case of POH-A_L_PEI, at pH below 2.5, sorption remains very low, below 0.1–0.13 mmol g^−1^ (but higher than for A_L_PEI). However, at pH above 2.5, a strong increase in sorption is observed, despite the overall positive charge of the sorbent (pH_PZC_: 6.82). The deprotonation limits the repulsion effect, but this is negligible compared to the strong contribution of new functional groups (phosphonic acid moieties) in the binding of REEs. The binding of metal ions occurs mainly through chelation on phosphonic groups although carboxylate groups (at pH above pH 4) may also contribute, together with some amine groups (limited number of free groups due to the high yield of grafting). Appendix A shows the semi-quantitative EDX analysis of the sorbent after metal sorption at pH 5. Cl content is not affected by metal sorption; the sorption of REEs is confirmed by the appearance of La (2.64% weight concentration, WC) and Tb (WC: 3.09%). Above pH 6, the formation of hydrolyzed species may cause partial precipitation of the metals. 

Appendix A reports the pH changes during metal sorption. It is noteworthy that the highest changes are observed for POH-A_L_PEI, and more specifically in the pH range 3–5 (maximum at pH 4). This is consistent with the titration profiles obtained in the pH-drift method; however, the pH variations are less marked in the presence of metal ions. It is also remarkable that, in the pH range 3–4, A_L_PEI slightly increases the pH, while a reciprocal trend is observed for POH-A_L_PEI. This is another indication of the change in the sorption mechanism for the functionalized sorbent: the release of protons from phosphonic groups (during the sorption of REEs) decreases the pH (up to 1 unit at pH_0_ 4).

#### 3.3.4. La (III) and Tb (III) Sorption Mechanisms onto POH-A_L_PEI

The combination of the information collected from FTIR and XPS data (shifts of wavenumbers, appearance and disappearance of signals (such as sulfate, OH, C-O and P-O), changes in the intensity of relevant XPS contributions or appearance of new signals such as S 2*p*) with acid–base titration (pH_PZC_) allows for proposing different mechanisms for metal sorption. These mechanisms may include:(a)Ion exchange of REE cation with Ca^2+^ bonded to carboxylic groups (resulting from ionotropic gelation, confirmed by Ca^2+^ disappearance on XPS spectra), and/or with protons from protonated amine groups and from hydroxyl groups (pH decrease after metal sorption).(b)Chelation mechanism on the electron doublet of nitrogen from amines groups, hydroxyls from polysaccharides, or from grafted phosphate groups.

The metal binding may occur through the uptake of trivalent metals or sulfate species (as shown by speciation diagram and both FTIR and XPS analyses). Scheme 2 shows the binding mechanisms potentially involved in metal binding. 

## 4. Conclusions and Perspectives

This preliminary study focuses on the design of a new functionalization method of algal biomass/polyethylenimine composite: the activation of the sorbent with epichlorohydrin followed by the phosphorylation of the activated material and its de-esterification. The chemical modifications are followed using FTIR and XPS spectroscopic methods not only to identify changes in chemical groups but also to interpret the binding mechanisms involved in the uptake of two rare earth elements in mild acidic solutions (pH close to 4). In combination with TGA characterization, elemental analysis, and titration (for the determination of pH_PZC_ values), it is possible to correlate chemical functionalization with remarkable sorption properties of the new sorbent. Sorption of rare earth elements (at least, partially in the form of sulfate complexes), which increases with pH, involves contributions from amine, hydroxyl, carboxylate and phosphonate groups (in relation with the multifunctional composition of the composite). FTIR analysis confirms the remarkable stability of the sorbent after being subjected to five cycles of sorption and desorption (using acidic calcium chloride solution). 

Under the experimental conditions selected for this study, the sorption capacity of the functionalized sorbent is increased 4.5 times for La (III) and up to 6.9 times for Tb (III), compared with raw support. This remarkable enhancement in sorption properties clearly justifies further investigation of the sorption properties of this new sorbent. A complementary study is currently being investigated focusing on the sorption of these two metal ions including uptake kinetics, sorption isotherms, metal desorption kinetics, effective recycling and selectivity in synthetic solutions. In this companion work, the sorbent is also considered as part of the global treatment of an Egyptian ore. This complementary study includes: acidic heap leaching, sorption on quaternary ammonium resin, sorption on POH-A_L_PEI, metal desorption from functionalized sorbent, oxalic acid precipitation for selective recovery of REEs and successive precipitations of the residue for separation of iron and aluminum and final recovery of uranium.

Sorbents loaded with rare earth elements (typically with lanthanum) have been successively used for the removal of phosphorous from aqueous effluents [70,71,72]. This may be an interesting field for the application of these new materials.

## Data Availability

Data are available from corresponding authors.

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
