# Peer review of "Synthesis of a New Phosphonate-Based Sorbent and Characterization of Its Interactions with Lanthanum (III) and Terbium (III)"

_polymers, 2021, doi:10.3390/polym13091513_

Round 1

Reviewer 1 Report

The authors synthesized a new phosphonate-based sorbent. The characterization such as XPS and FTIR is conducted to investigate the synthesis procedure. The excellent cyclability of repeatability was demonstrated. I suggest that this manuscript can be accepted after addressing the below questions.

1-What is the maximum absorption value of this current sorbent?

2-What is the influence from the pH value?

3-Any influence from other ions in the tested solution?

4-Mark the sub-figure using (a), (b), (c), (d).

5- Improve the figure quality such as Figure 6.

6- Some related references should be added in this area.

Author Response

Please, see the enclosed file

Reviewer 2 Report

I have read the Manuscript Polymers-1202753 titles “Synthesis of a new phosphonate-based sorbent and characterization of its interactions with lanthanum(III) and terbium(III)” having as corresponding authors Prof. Mohammed F. Hamza and Enrique Rodríguez-Castellón.

In this work the authors prepared beads composite from alginate and PEI. Through various chemical modifications the beads contains phosphonate groups that improves the capacity as sorbents of REE.

This is a well-prepared and discussed manuscript and I recommend for publication in Polymers.

There are a few details in the manuscript. Below are listed:

Throughout the manuscript: it is recommended to homogenize the style of the samples, especially emphasizing the subscripts, to avoid confusion (as an example, Scheme 2, Figure S1 etc.).

In the materials part, please provide the molecular weight for Polyethylene glycol diglycidyl ether (line 94).

Is this scheme missing in the manuscript maybe?

Scheme 1. summarizes the different steps of the synthesis procedure

Please correct as Scheme 1.

Scheme 2. Procedure for the synthesis of POH-ALPEI beads (line 150)

In the supplementary information:

Figure S1. In b) there is a mistake in POH-ALPEI. Please correct.

Minor details:

Please homogenize has polyethylenimine

In the keywords:

Polyethyleneimine

Polyethyleneimine (line 62)

Polyethyleneimine (line 217)

Polyethyleneimine (line 554)

In the conclusion

Please correct: idnetify (line 557)

Author Response

Please, see the enclosed file

Reviewer 3 Report

This paper illustrated “Synthesis of a new phosphonate-based sorbent and characterization of its interactions with lanthanum (III) and terbium (III)”. The results are interesting, but this paper is a major revision needed.

The comments are listed below.

  1. Keywords are too long, should be shortened.
  2. Lines 57: Please check the space between “particularly” and “important”. Please also recheck the whole manuscript.
  3. Lines 143~149: Please rearrange the Scheme 1. There are obvious layout errors.
  4. Lines 356, 446, 453, 457 and 499: Please rearrange the Figures 2, 5, 6 and 8. There are obvious layout errors.
  5. Lines 548~551: There is an obvious layout error. It should be “Scheme 3”.
  6. It is suggested that the author should explore the effect of changing the amount of the components on the metal sorption/desorption properties.
  7. It is suggested that the author should show SEM images of sorbent in the manuscript.

Author Response

Please, see the enclosed file

Round 2

Reviewer 3 Report

The authors have revised the manuscript according to the comments and suggestions. This paper is ready for publication.